# Microstructure and Mechanical Properties of AlN/Al Joints Brazing by a Sputtering Al/Cu Bilayer Film Solder

**DOI:** 10.3390/ma15072674

**Published:** 2022-04-05

**Authors:** Zhun Ran, Hailong Shang, Bingyang Ma, Rongbin Li, Fujun Shangguan, Dayi Yu

**Affiliations:** School of Materials Science and Engineering, Shanghai Dianji University, Shanghai 201306, China; ranhuanyu1998@163.com (Z.R.); maby@sdju.edu.cn (B.M.); lirb@sdju.edu.cn (R.L.); fjshangguan@163.com (F.S.); woshitr_ydy@163.com (D.Y.)

**Keywords:** AlN ceramic, Al metal, brazing, wettability, shear strength

## Abstract

This paper presents a magnetron sputtering Al/Cu bilayer film solder to realize the brazing of AlN ceramic and Al metal. The effect of different temperatures on the structure and mechanical properties of brazed joints is studied. The results show that the sputtered Al particles have a sputtering wetting effect on AlN ceramic. The contact angle of molten Al on AlN ceramic with Al film deposited at 700 °C is as low as about 30°. While the contact angle of molten Al on AlN ceramic without Al film deposited at 1000 °C is about 89°. There is a large amount of Cu enrichment in brazed joints at 600 °C. The weld structure is a mixture of Al solid solution and AlCu compound. The shear strength of the brazed joint is only 70.6 MPa, and the joint fracture shows a large number of brittle fracture morphologies. With the increase of brazing temperature, the phenomenon of Cu enrichment in the joint gradually weakens, and the weld structure gradually transforms into a solid solution of Cu in Al. The shear strength of the brazed joint continues to increase, and the joint fracture morphology gradually changes from brittle fracture to furrow-like plastic fracture morphology. When the brazing temperature is increased to 660 °C, the distribution of Cu in the joint is evenly dispersed, and the shear strength of the brazed joint reaches the highest value of 107.8 MPa. The joint fracture is completely furrow-like plastic fracture morphological composition.

## 1. Introduction

The application of high-power semiconductor devices, such as insulated gate bipolar transistors (IGBT), silicon-controlled rectifiers (SCR), power transistors, is becoming more and more extensive. Due to the excellent reliable insulation, low dielectric constant and dielectric loss, AlN ceramic substrate has a wide range of applications in the field of microelectronics and electronic packaging such as large-scale integrated circuits and high-power components. Limited by poor thermal conductivity, the substrate needs to be brazed with metal to ensure the high-power density and excellent heat dissipation performance of the device, such as directly bonded copper/ceramic (DBC) substrate and directly bonded aluminum/ceramic (DBA) substrate [1,2,3]. Comparing with directly bonded copper/ceramic (DBC) substrate, directly bonded aluminum/ceramic (DBA) has more stably thermal cycling reliability, more excellent thermal fatigue resistance, thermal stability, and low weight, thus leading to longer life-time devices.

The good wettability of the liquid solder to AlN ceramic is a prerequisite for the realization of AlN/Al brazing connection. The traditional methods usually coat the active metal layer on AlN [4,5] or add active elements to the solder [1,6] to achieve wetting by reacting with AlN ceramic. Both methods create a reaction layer of active metal and ceramics to improve the wettability of liquid metal on ceramics. For example, Terasaki et al. [4] coated an Ag-Ti layer on the surface of AlN ceramic, and then brazed the AlN ceramic and Cu plates. The presence of TiN reactive layers and CuTi intermetallic compounds at the interface was observed. Sivaprahasam et al. [6] used AgCuTi active solder to realize the brazing connection of AlN/Cu and AlN/Ni.

However, these reactive wetting brazing methods will produce a reactive transition layer at the AlN ceramic interface, which not only reduces the mechanical properties of the joint, but also hinders heat conduction, thereby reducing the heat dissipation performance of the substrate.

Different from other active solders, Al is a rare metal that can directly wet AlN ceramic without reaction, which makes it a very promising solder. However, this wetting temperature is as high as 850 °C [7], and some studies believe that the wetting temperature of AlN/Al even exceeds 1000 °C [8], while the melting point of pure Al is only about 660 °C. How to coordinate the contradiction between the low melting point of pure Al and the high wetting temperature of AlN/Al is one of the key points to realize AlN/Al brazed by Al solder. On the other hand, there is a very dense and stable oxide film on the Al surface. How to remove this oxide film and achieve direct contact between the liquid solder and the base metal is another key point.

In order to solve the above problems and to realize the non-reactive transition layer brazing of AlN ceramic and Al metal using Al alloy as solder. In this paper, the magnetron sputtering method is used to deposit Al/Cu bilayer alloy film on the surface of AlN ceramic and Al as a solder to realize the brazing connection of AlN ceramic and Al metal. Furthermore, the effects of different brazing temperatures on the microstructure and mechanical properties of AlN/Al brazed joints are studied.

## 2. Materials and Methods

### 2.1. Experimental Design

In order to realize the brazing of AlN ceramic and Al metal, the Al/Cu bilayer films are deposited on the surface of AlN ceramic and Al as solder. On the one hand, the high kinetic energy of sputtering particles is used to realize the “sputtering wetting” of Al to AlN ceramic at a lower temperature. On the other hand, by using the low melting point of Al/Cu alloy film and the “local melting” method of partial melting of solder, the brazing connection between AlN ceramic and Al metal is realized. The schematic diagram of brazing is shown in Figure 1.

The purity of the AlN ceramic and the Al metal substrates used in this paper are 99% and 99.9%, respectively. Both of them measure 2 mm × 25 mm × 25 mm.

The Al/Cu bilayer film solder in Figure 1 is prepared by ANELVA SPC-350 multi-target magnetron sputtering. The polished AlN ceramic and Al substrates are ultrasonically cleaned in alcohol and placed in the vacuum chamber of the multi-target magnetron sputtering system. When the background vacuum of the vacuum chamber reaches 5.0 × 10^−4^ Pa, Ar gas with a purity of 99.999% is filled into the vacuum chamber and the working pressure is maintained at 0.6 Pa. The Ø 76 mm Al target (purity 99.99%) and Cu target (purity 99.99%) is controlled by DC and RF cathode, respectively. During the experiment, a 10 μm Al film is deposited on the surface of the AlN ceramic and Al metal, followed by 0.5 μm Cu film. The substrate is neither heated nor applied with a negative bias.

### 2.2. Wetting Experiment

In order to reveal the “sputtering wetting” effect of magnetron sputtering Al film on AlN ceramic, this paper designed two group of AlN ceramic samples without (Figure 2a) and with (Figure 2b) Al film deposited on the surface. The “sputtering wetting” effect is revealed by comparing the wetting behavior of molten Al on two groups of AlN ceramic samples.

The wettability experiment is carried out using a sealed chamber method that can cut off the oxygen and nitrogen sources [9]. The structure of the sealing chamber is shown in Figure 3. A flat-mouth quartz cup is placed upside down on a quartz plate, where aluminum base solder is placed between them. A stainless-steel weight is pressed on the top of the quartz cup. The AlN ceramic sample to be observed is placed in the quartz cup, with a pure aluminum block in the center of its surface. This assembly is placed in a vacuum heating furnace. The heating furnace is vacuumed to 10^−1^ Pa, and then heated until the aluminum base solder sheet is melted to braze the quartz cup and the quartz plate together through the liquid solder. A quartz sealed chamber that does not exchange gases with outside environment is prepared. This assembly cuts off the access of O_2_ and N_2_ and obtains a very low oxygen partial pressure and nitrogen partial pressure sealing environment.

In the wetting experiment, the AlN ceramic without and with Al film are placed in the sealed chamber, heated to 1000 °C and 700 °C, respectively, then cooled in the furnace. The wetting behavior is revealed by comparing the contact angle of molten Al on these two kinds of AlN ceramics.

### 2.3. Brazing Experiment

The AlN ceramic and Al metal are brazed by vacuum brazing. The AlN ceramic and Al with Al/Cu bilayer film solder deposited on the surface are placed opposite to each other in a vacuum furnace (Figure 1a). A small stainless-steel block is placed on the top of the sample for pressure and fixation. After pumping the vacuum to 10^−1^ Pa, the samples are brazed at 600 °C, 620 °C, 640 °C and 660 °C, respectively. During the brazing process, the temperature rise rate of the vacuum furnace is 10 °C/min, the holding time is 2 h, and the samples are cooled with the furnace after welding.

### 2.4. Characterization

The Hitachi S-3400N scanning electron microscope (SEM) and its affiliated Apolloxp X-ray energy dispersive spectroscopy (EDS) are used to observe the structure and morphology of the brazed joints and the shear fractures of the joints. The shear strength of the brazed joint is tested using an electronic tensile testing machine and a special fixture (Figure 4). The loading speed is 0.3 mm/min, and the joint is tested with 10 samples (interface size 3 mm × 2 mm) and averaged value.

## 3. Results

### 3.1. Wetting Behavior of Sputtering Al Particles on AlN

The Figure 5 shows the contact angle of molten Al on the surface of two groups of AlN ceramics. It can be seen from the figure that both two groups of Al blocks turn into balls, which indicates that the sealing chamber designed in this experiment avoids the influence of air on the wetting experiment. By comparing the contact angles, it is found that the contact angle of AlN ceramic without Al film is 89° at 1000 °C, while that of AlN ceramic with Al film is only 30.2° at 700 °C, showing excellent wetting effect.

### 3.2. Microstructure Characteristics of AlN/Al Joint

Figure 6 shows SEM images of AlN/Al joints at different temperatures and the Al, Cu, N element scanning images of the corresponding area. It can be seen from Figure 6 that the interface of brazed joint at 600 °C, 620 °C, 640 °C, 660 °C has no porosity and other brazing defects. The brazed joint presents dense and full as cast structure. The filler metal can achieve good metallurgical bonding with AlN ceramic and Al, which indicates that the wetting of AlN ceramic by Al has been significantly improved at various temperatures.

The element scanning of the weld at 600 °C (Figure 6a) found that there are a large number of Cu-rich areas in the weld. At this time, the enriched area is mainly AlCu eutectic structure. It shows that due to the lower brazing temperature, the “local melting” area is narrower, and the Cu content in the weld is relatively high. As the brazing temperature is increased to 620 °C (Figure 6b), although some Cu-enriched areas can still be observed in the brazed joint, they have been greatly reduced compared with 600 °C, indicating that increasing the temperature makes the Cu-enriched areas gradually melt and gradually diffuse to the surroundings. When the temperature is further increased to 640 °C (Figure 6c), the Cu element has been highly dispersed in the brazing seam. The microstructure of the weld is dominated by the solid solution of Cu in Al. Continue to increase the temperature to 660 °C (Figure 6d), and the Cu element in the weld has been completely dispersed.

### 3.3. Mechanical Properties of AlN/Al Joint

The Figure 7 shows the shear strength of AlN/Al joints at 600 °C, 620 °C, 640 °C, 660 °C. The shear strength of brazed joints at 600 °C is 70.6 MPa. As the temperature increases, the shear strength of the joints increases and reach the maximum shear strength of 107.8 MPa at 660 °C.

Figure 8 shows SEM images of the shear fractures of each brazed joint at 600 °C, 620 °C, 640 °C, 660 °C. The shear fracture at 600 °C (Figure 8a) shows the brittle fracture morphology. The composition analysis results show that there is a large area of Cu enrichment at this temperature. This brittle AlCu compound makes the combination of AlN ceramic and Al at 600 °C inferior, and has low shear strength. The brittle fracture structure at 620 °C (Figure 8b) has been obviously reduced, and the plough shaped scratch plastic fracture morphology of Al caused by shear can be seen in the local area. The surface scanning results show that the large area of Cu enrichment in the joint fracture has disappeared, and it exists in the form of small and dispersed Cu enrichment area, which also proves that increasing temperature can promote the diffusion of Cu. The shear strength of the joint is further improved. At 640 °C (Figure 8c), the fracture morphology of the brazed joint showed a lot of furrow-like plasticity morphology. The scanning results of the element surface shows that the Cu element in the brazed joint is evenly dispersed, and almost no enrichment. The weld is mainly in the form of a solid solution structure of Cu in Al, which greatly improves the plasticity of the weld. With the further increase of the brazing temperature to 660 °C (Figure 8d), the morphology of the brazed joints all transformed into a furrow shape.

## 4. Discussion

### 4.1. Sputtering Wetting Effect

A large number of wetting experiment results [7,8,10,11,12,13,14] show that temperature is an important factor in improving the wettability of Al/AlN system. The improvement of this wetting state mainly comes from the reduction of the liquid/solid interfacial tension of the system. Its essence is that the Al atoms in the melt and the N atoms on the AlN ceramic surface gradually form an Al-N chemical bond shared by electrons [15]. The formation of the Al-N chemical bond significantly reduces the interfacial tension between the Al liquid and the AlN ceramic. The increase in temperature provides the kinetic energy for the Al atoms in the solution to overcome the potential barrier necessary for the formation of the chemical bond. This is also confirmed by the results in Figure 5a that the contact angle of Al on the AlN ceramic sample without Al film deposited at 1000 °C is reduced to below 90°. Meanwhile, the Al film sample deposited in Figure 5b shows a phenomenon that sputtered Al film can improve the wetting state. As we all know, during magnetron sputtering, the kinetic energy of gas phase particles (atoms or ions) sputtered from the cathode surface are as high as 10^0^ eV [16], which is much higher than the thermal kinetic energy of molten aluminum atoms. When these high-energy sputtered Al particles are directly deposited, the impact on the AlN ceramic surface is sufficient to overcome the energy barrier, and form the Al-N chemical bond with N atoms on the AlN ceramic surface. Once the Al-N chemical bonds are formed, they can still be maintained after the Al film is heated and melted, so that the Al liquid can wet the ceramic at the just melting temperature. Therefore, although the sputtered Al thin film is solid, they have “wetted” AlN ceramic. This phenomenon can be called “sputtering wetting”. In fact, it can be seen from the above analysis that as long as Al atoms have sufficiently high energy, they can form Al-N chemical bonds with wetting effect on the interface with AlN ceramic, regardless of whether the interface is a liquid/solid interface or a solid/solid interface. This “sputtering wetting” effect provides space for brazing to connect AlN ceramic and Al metal at a lower temperature.

### 4.2. Removal of Oxide Film and Local Melting Brazing

The effective removal of surface oxide film has always been a problem that Al brazing (including brazing of Al materials and brazing with Al solder) must be faced. As we all know, the surface of Al liquid has dense and stable Al_2_O_3_ solid oxide film. The melting point of this oxide film is as high as 2050 °C, and its chemical stability is extremely high. Even at a high temperature of 1000 °C, it does not decompose till the oxygen partial pressure drops to 10^−30^ Pa [17]. This solid and dense oxide film prevents the direct contact between the Al molten and the AlN ceramic. The existing methods mainly use chloride flux or magnesium vapor to remove molten Al and oxide film. Compared with the traditional methods, the magnetron sputtering Al/Cu bilayer film solder used in this paper can remove the oxide film more cleanly and conveniently. The Al layer is directly deposited on the AlN ceramic surface, which not only realizes the Al direct contact with AlN ceramic, high-energy sputtered Al particles also form a chemical combination of Al-N bonds with N on the AlN ceramic surface, forming a “sputtering wetting” effect. The Cu coating on the surface can prevent the formation of new oxide film on the surface of Al film. The loose oxides (Cu_2_O, CuO) generated on the surface will also react with the Al liquid after the Al layer is melted.

In addition, the design of the Al/Cu bilayer film solder can reduce the melting point of the Al solder, making it possible to braze AlN ceramic and Al metal by the “local melting” method in which the solder is partially melted in this article. The microstructure and mechanical properties of the brazed joints are improved by controlling the diffusion of Cu with the change of brazing temperature. Combined with the experimental results in this article, the weld seam in the brazed joint at 600 °C is relatively narrow. There are a lot of brittle AlCu compound phases in the weld seam, which makes the shear strength relatively low. The weld fracture showed the morphological characteristics of brittle fracture. As the temperature increases, the diffusion capacity of Cu increases, and the content of the brittle AlCu compound phase decreases. The weld gradually transforms into a solid solution of Cu in Al. The shear strength is obviously improved, and the furrow-like plastic fracture morphology area in the brazing fracture is gradually increased.

## 5. Conclusions

In this study, Al-Cu alloy brazing filler metal was used to successfully realize the brazing of AlN/Al. The following conclusions are reached:(1)Sputtered Al particles have a sputtering wetting effect on AlN ceramic. The contact angle of molten Al on AlN ceramic with Al film deposited at 700 °C can be as low as about 30°. While the contact angle of molten Al on AlN ceramic without Al film deposited is only 89° at a high temperature of 1000 °C.(2)The direct brazing of AlN/Al is realized using sputtered Al/Cu bilayer film solder. The influence of Al surface oxide film on brazing is effectively eliminated. The joints brazed at different temperature are all dense and full, and there is no reaction transition layer at the interface.(3)The shear strength of the joint brazed at 600 °C is 70.6 MPa, the weld structure is a mixture of Al solid solution and AlCu compound, and the joint fracture has a brittle fracture morphology. With the increase of brazing temperature, the shear strength of the brazed joint continues to increase. The weld structure gradually transforms into a solid solution of Cu in Al. The joint fracture morphology gradually changes from brittle fracture to furrow-like plastic fracture morphology. When the brazing temperature is increased to 660 °C, the distribution of Cu in the joint is evenly dispersed, and the shear strength of the brazed joint reaches the highest value of 107.8 MPa. The joint fracture is completely furrow-like plastic fracture morphological composition.

## Figures and Tables

**Figure 1 materials-15-02674-f001:**
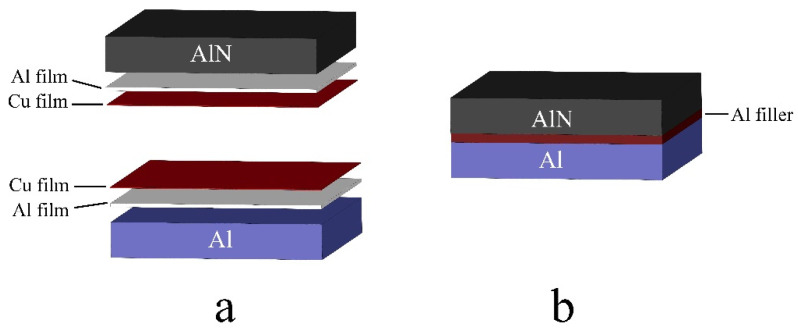
Schematic diagram of AlN/Al brazing. (**a**) Schematic diagram of film deposition; (**b**) Schematic diagram of brazing.

**Figure 2 materials-15-02674-f002:**
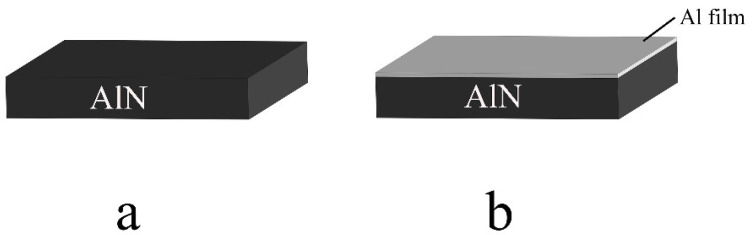
Two groups of AlN ceramic samples (**a**) without Al film deposited; (**b**) with Al film deposited.

**Figure 3 materials-15-02674-f003:**
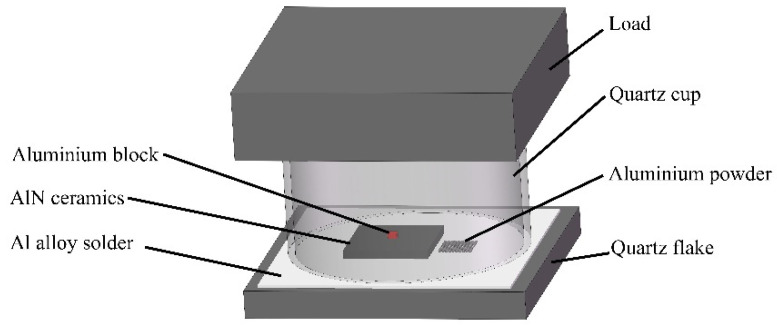
Schematic diagram of the sealed chamber structure.

**Figure 4 materials-15-02674-f004:**
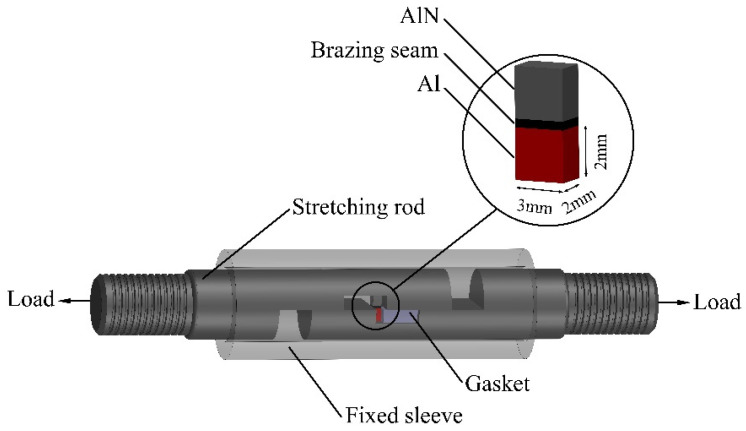
Schematic diagram of the shear strength fixture for AlN/Al brazed joints.

**Figure 5 materials-15-02674-f005:**
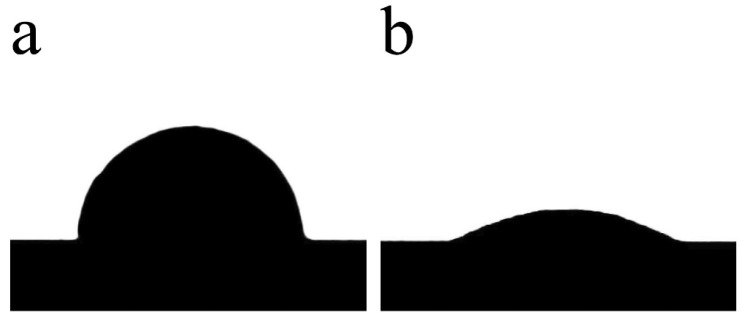
Wetting behavior of molten Al on two samples. (**a**) without film at 1000 °C; (**b**) with Al film at 700 °C.

**Figure 6 materials-15-02674-f006:**
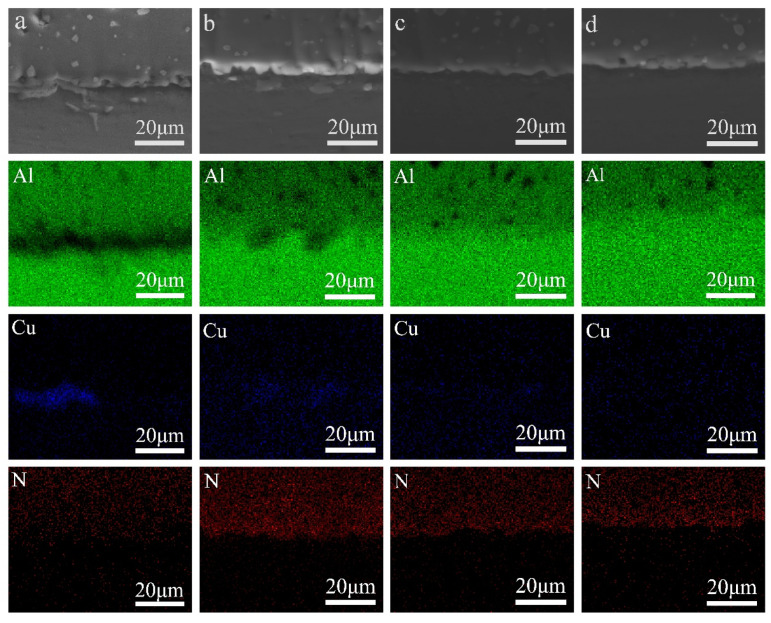
Scanning electron microscope images and scanning results of Al, Cu and N elements of brazed joints at different temperatures (**a**) 600 °C; (**b**) 620 °C; (**c**) 640 °C; (**d**) 660 °C.

**Figure 7 materials-15-02674-f007:**
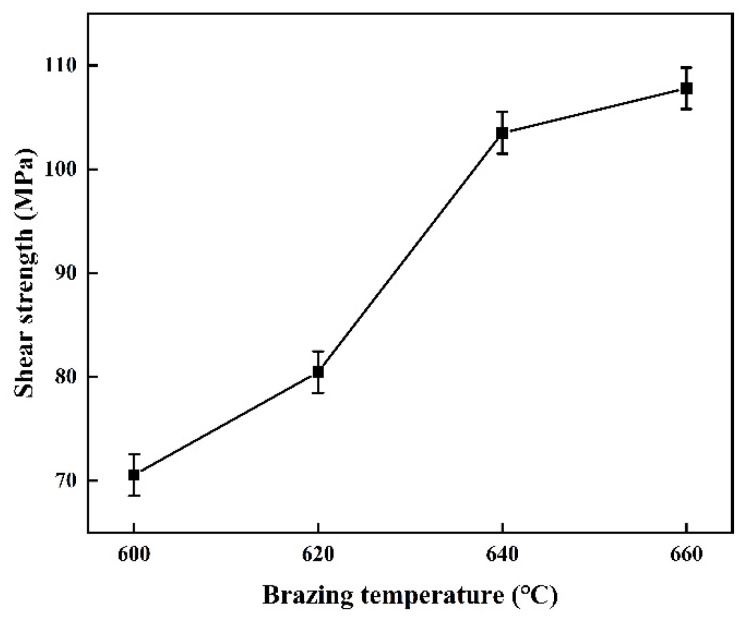
Shear strength of AlN/Al brazed joints. The brazing temperatures of the joints are 600 °C, 620 °C, 640 °C and 660 °C.

**Figure 8 materials-15-02674-f008:**
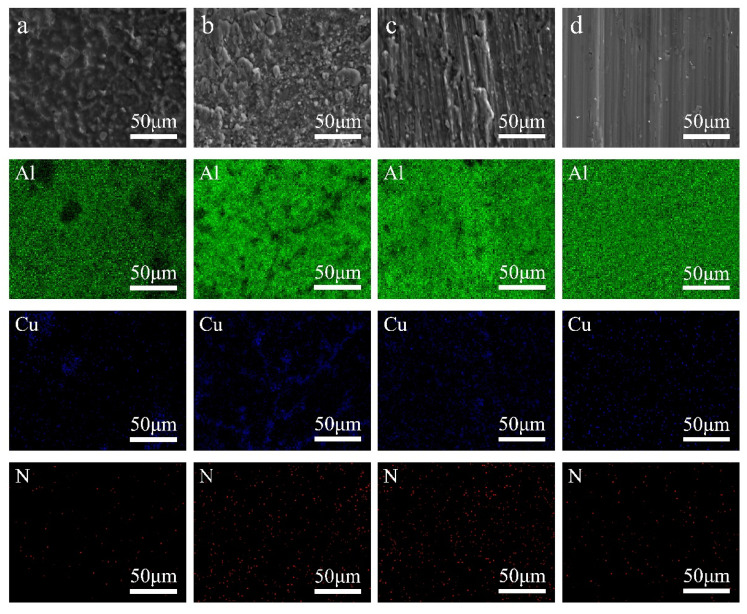
SEM images of shear fracture and scanning results of Al, Cu and N elements of brazed joints at different temperatures. (**a**) 600 °C; (**b**) 620 °C; (**c**) 640 °C; (**d**) 660 °C.

## Data Availability

The data presented in this study are available on request from the corresponding author.

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
