# Peer review of "Microstructure and Mechanical Properties of AlN/Al Joints Brazing by a Sputtering Al/Cu Bilayer Film Solder"

_materials, 2022, doi:10.3390/ma15072674_

Round 1
Reviewer 1 Report
The present manuscript described the joints formed by solder for aluminum nitride ceramic materials. The abstract was written ok. However, the introduction was not comprehensive. It did not provide the justification for the need for this research. The experimental procedure is clear. It was concluded that as the temperature increases the joint fracture transform from brittle failure to plastic failure. the conclusion was drawn based on the experimental observation and the manuscript deserves to be published. However, the following needs to be addressed before publication:
Abstract: Please rewrite the abstract in chronological order of increasing temperature. Lines 13-14: To be consistent, please remove the .0 after 89.0 (20°, 89°). Again, please be consistent with your material (AlN ceramic vs only AlN). Also, the joint strength was 70.6 MPa at 600 °C, and the strength increased to 107.8 MPa at 660 °C. What was the joint strength at 1000 °C? What was the contact angle at 660 °C?
Introduction: The literature needs to be thoroughly studied to justify the need for this research? What was the main goal of this research? Was it the magnetron sputtering method or the effect of different brazing temperatures? The novelty of the current research should be established by a thorough literature review which developed the need for the research by a logical sequence of arguments backed by reference. lines 58-61: it appears suddenly what will be studied in this manuscript without preparing the reader properly.
Experimental: It was not clear what was the starting material. Was it AlN with al film and then coated with al film? Which finally coated with Cu film by magnetron sputtering? Figure 2 is unnecessary. Why were only 1000 °C and 700 °C chosen to investigate the wettability? What will happen if you change the temperature? It is not ok to conclude anything based on just 2 points (1000 °C and 700 °C). The joints were made at four different temperatures (600 °C; 620 °C; 640 °C; 660 159 °C). However, this information was only available in the result section at line 159. This information should be provided in the experimental section.
Results: Figure 6 shows the SEM images for AlN/Al joints at 600-660 °C. Is it for AlN ceramic with Al coating and then Cu coating by sputtering? The same for other results. The results section is completely confusing it does not communicate well with the experimental section.
The paper needs a massive restructure before publication.
Author Response
Response to the Reviewers
The authors would like to thank the editor and reviewers for the valuable comments and suggestions. Per your suggestions, the manuscript has been carefully revised. The following is the point by point response to the reviewers’ comments. Please note that all reviewer comments are shown in times new roman, while our response italics.

Reviewer 2 Report
Dear Authors,
The research topics you have conducted are interesting and important in the context of the production of electronic components. The content of the article does not raise my doubts in terms of content and science. Nevertheless, in order to improve the quality of publications, some issues should be clarified more widely and some corrections made. I propose to refer to the following issues in the body of the article:
- Explain how AlN ceramics and Al metal were prepared for the surface.
- Please specify the heating conditions of the joined elements and the cooling of the brazed joint (heating and cooling speed).
- In the research presented, surface wetting is a fundamental phenomenon. In the article, you refer to the contact angle. The graphics shown in Figure 5 do not convince me that this parameter has been measured. By what method was the value of the wetting angle determined. Whether the surface tension of the liquid metal was measured. This issue must be explained and described in detail.
- Specify the standard deviation value in Figure 7. The caption under the drawing may indicate that the shear strength has been tested at elevated temperatures. Describe the X-axis in more detail and correct the caption under the figure.
- In the article, it is also worth indicating the specific use (applicability) of the process examined.
- It is also worth considering the linguistic correction of the article.
Yours sincerely,
Reviewer
Author Response

(The authors gave the same response as above.)

Round 2
Reviewer 1 Report
The author made significant efforts to improve the manuscript by addressing the reviewers' comments. It can be accepted now.